# Activation of the β-adrenergic receptor exacerbates lipopolysaccharide-induced wasting of skeletal muscle cells by increasing interleukin-6 production

**Shino Matsukawa, Shinichi Kai[ID]\*, Hideya Seo, Kengo Suzuki, Kazuhiko Fukuda**

Department of Anesthesia, Kyoto University Hospital, Kyoto, Japan

\* s.kai0627@gmail.com

**Data Availability Statement:** All relevant data are within the paper and its Supporting Information files.

## Abstract

The skeletal muscle mass has been shown to be affected by catecholamines, such as epinephrine (Epi), norepinephrine (NE), and isoproterenol (ISO). On the other hand, lipopolysaccharide (LPS), one of the causative substances of sepsis, induces muscle wasting via toll-like receptors expressed in skeletal muscle. Although catecholamines are frequently administered to critically ill patients, it is still incompletely understood how these drugs affect skeletal muscle during critical illness, including sepsis. Herein, we examined the direct effects of catecholamines on LPS-induced skeletal muscle wasting using the C2C12 myoblast cell line. Muscle wasting induced by catecholamines and/or LPS was analyzed by the use of the differentiated C2C12 myotubes, and its underlying mechanism was explored by immunoblotting analysis, quantitative reverse transcription polymerase chain reaction (qRT-PCR), enzyme-linked immunosorbent assay (ELISA), and the TransAM kit for p-65 NF-κB. Epi augmented myosin heavy chain (MHC) protein loss and reduction of the myotube diameter induced by LPS. LPS induced C/EBPδ protein, Atrogin-1 and inteleukin-6 (IL-6), and these responses were potentiated by Epi. An IL-6 inhibitor, LMT28, suppressed the potentiating effect of Epi on the LPS-induced responses. NF-κB activity was induced by LPS, but was not affected by Epi and recombinant IL-6, and the NF-κB inhibitor, Bay 11–7082, abolished Atrogin-1 mRNA expression induced by LPS with or without Epi. NE and ISO also potentiated LPS-induced IL-6 and Atrogin-1 mRNA expression. Carvedilol, a nonselective β-adrenergic receptor antagonist, suppressed the facilitating effects of Epi on the Atrogin-1 mRNA induction by LPS, and abolished the effects of Epi on the MHC protein loss in the presence of LPS. It was concluded that Epi activates the β-adrenergic receptors in C2C12 myotubes and the IL-6-STAT3 pathway, leading to the augmentation of LPS-induced activation of the NF-κB- C/EBPδ-Atrogin-1 pathway and to the exacerbation of myotube wasting.

**Funding:** The study was supported by Grants-in-Aid for Young Scientists from the Japan Society for the Promotion of Science (grant numbers 18K16447 to SK, and 17K16729 to HS), and Research Grants from The Nakatomi Foundation (203180700108) (https://www.nakatomi.or.jp/en/) to SK. The funders had no role in study design, data collection and analysis, decision to publish, or preparation of the manuscript.

**Competing interests:** The authors have declared that no competing interests exist.

## Introduction

Catecholamines, such as epinephrine (Epi), norepinephrine (NE), and isoproterenol (ISO), produce cellular effects through the adrenergic receptors expressed in a variety of tissues, including blood vessels, heart, adipose tissue, and skeletal muscle. Skeletal muscle predominantly expresses the β-adrenergic receptors, especially the $\beta_2$-subtype [1]. Although several studies reported that stimulation of the β-receptors increased skeletal muscle mass [2], other studies have demonstrated that the blockade of the β receptor is beneficial for protein proteolysis and energy expenditure in skeletal muscle in sepsis [3, 4]. Furthermore, a recent clinical study reported that catecholamine may be one of the risk factors in the development of muscle wasting [5], which constitutes a serious clinical consequence associated with critical illness that results in significant morbidity and mortality [6, 7]. However, it remains to be clarified whether catecholamines can affect skeletal muscle mass in systemic inflammation that constitutes a common problem in critically ill patients.

Mechanisms for skeletal muscle wasting have been explored from multiple aspects. Proteolysis in skeletal muscle was shown to be increased owing to the upregulation of the ubiquitin proteasomal pathway (UPP) in several animal sepsis models, including the administration of lipopolysaccharides (LPSs) [8, 9]. It was also reported that LPS induces wasting in myotubes differentiated from a mouse myoblast cell line C2C12 by upregulating UPP via toll-like receptors [10]. Furthermore, it was shown that two muscle-specific E3 ubiquitin ligases, Atrogin-1 and MuRF1, function as essential regulators of UPP, and the increase in their expressions precedes the loss of muscle weight [11]. On the other hand, the STAT3 pathway and NF-κB activation play critical roles in muscle wasting [12]. It was demonstrated that the conditioned media from colon and lung cancer cells induce STAT3 activation and CCAAT/enhancer-binding protein (C/EBP) δ expression in C2C12 cells, leading to an increase in Atrogin-1 expression [13]. C/EBPδ is a transcription factor that mediates stress response and is induced by LPS through NF-κB signaling [14]. There is abundant experimental evidence that proinflammatory cytokines released by innate immune cells and skeletal muscle cells in response to various stresses [15], such as interleukin-6 (IL-6) and tumor necrosis factor-α (TNF-α) in plasma levels which are often elevated in septic patients [16, 17], are involved in muscle wasting [18].

In the present study, we performed in vitro experiments using C2C12 myotubes to evaluate the effects of catecholamines on LPS-induced muscle wasting. We demonstrated that Epi exacerbated LPS-induced myotube wasting by potentiating the NF-κB-mediated activation of the C/EBPδ-Atrogin-1 pathway via the IL-6-STAT3 pathway through activation of the β-adrenergic receptors.

## Materials and methods

### Cell culture

C2C12 myoblast (ECACC, Salisbury, UK) was maintained in Dulbecco's modified eagle medium (DMEM) that contained 10% fetal bovine serum at 37˚C in 5% $CO_2$. After the cells reached confluence, differentiation was induced by growing the cells in DMEM supplemented with 2% horse serum for 4 days.

### Immunoblotting assays

Whole-cell lysates were prepared as described previously [19]. Briefly, whole-cell lysates were prepared by using ice-cold lysis buffer [0.1% SDS, 1% Nonidet P-40 (NP-40), 5 mM EDTA, 150 mM NaCl, 50 mM Tris-Cl (pH 8.0), 2 mM DTT, 1 mM sodium orthovanadate, complete protease inhibitor$^{TM}$ (Roche Diagnostics, Tokyo, Japan) and 1:100 phosphatase inhibitor

cocktail (Nacalai Tesque, Kyoto, Japan) based on a protocol described previously [20]. Samples were centrifuged at 11,000 g to form a pellet from the cell debris. An equal amount of protein was fractionated by SDS-PAGE and subjected to an immunoblotting assay. Primary antibodies to phospho-STAT3 (T705) (1:2000; #9145), total-STAT3 (1:1000; #9132), and C/EBPδ (1:1,000; #2318), phospho-p70 S6K (The 389) (1:2000; #9205), total-p70 S6K (1:1000; #9202), phospho-4E-BP1 (The37/46) (1:2000; #2855), total-4E-BP (1:1000; #9644), phospho-CREB (Ser 133) (1:2000; #9198), total-CREB (1:1000; 9197S), were from Cell Signaling Technology (Beverly City, MA, USA). The antibody for myosin heavy chain (MHC) (1:4,000) was from R and D Systems (Minneapolis, MN, USA). Anti-β-actin mouse monoclonal antibody (Sigma-Aldrich, St Louis, MO, USA) was used as a control at 1:4000 dilution. Horseradish peroxidase-conjugated to sheep anti-mouse or donkey anti-rabbit IgG (GE Healthcare, Piscataway, NJ, USA) was used as a secondary antibody at a dilution of 1:2000. The signal was developed with enhanced chemiluminescence reagent (GE Healthcare). Experiments were repeated at least thrice. The intensity of each band was quantified with the software Image J (NIH, Bethesda, MD, USA) and used for subsequent statistical analyses.

## Quantitative reverse transcription polymerase chain reaction (qRT-PCR) analysis

First-strand synthesis and RT-PCR were performed using the one-step SYBR PrimeScript RT-PCR kit II (TAKARA, Ohtsu, Japan), according to the manufacturer's protocol. Amplification and detection were performed using the Applied Biosystems 7300 Real-time PCR system (Applied Biosystems, Foster City, CA, USA). The change in expression of each target messenger ribonucleic acid (mRNA) was calculated relative to the level of 18S ribosomal RNA (rRNA).

Primers for detection of mRNA are as follows:

Atrogin-1 Forward: 5′—ATGCACACTGGTGCAGAGAG –3′,
Reverse: 5′ -TGTAAGCACACAGGCAGGTC –3′,
MuRF1 Forward: 5′—CACGAAGACGAGAAGATCAACATC –3′,
Reverse: 5′—AGCCCCAAACACCTTGCA –3′,
IL-6 Forward: 5′—ACAACCACGGCCTTCCCTACTT –3′,
Reverse: 5′—CACGATTTCCCAGAGAACATGTG –3′,
TNF-α Forward: 5′—TCAACAACTACTCAGAAACAC –3′,
Reverse: 5′—AGAACTCAGGAATGGACAT –3′,
IL-1β Forward: 5′—GTGATATTCTCCATGAGCTTTG –3′,
Reverse: 5′—TCTTCTTTGGGTATTGCTTG –3′,
18 S Forward: 5′—GTAACCCGTTGAACCCCATT –3′,
Reverse: 5′—CCATCCAATCGGTAGTAGCG –3′.

## Enzyme-linked immunosorbent assay (ELISA)

Conditioned media from C2C12 cells were collected after each treatment. Mouse IL-6 in conditioned media was measured using the Mouse ELISA kit (cat. No. M6000B, R&D Systems) according to the manufacturer's protocol.

## Assays of transcription factor activity

To determine the amount of activated p-65 NF-κB, protein extracts were prepared using a nuclear extract kit (cat. No. 40010, ActiveMotif, Rixensart, Belgium). The aqueous amount of protein content was used for transcription factor activity analysis with TransAM kits for p-65 NF-κB (cat. No. 40097, ActiveMotif) according to the manufacturer's protocol. The amount of

active transcription factor was determined by measuring its binding capacity to a consensus binding sequence to assess the specificity of deoxyribonucleic acid (DNA) binding. Data are provided as relative units.

## Immunofluorescence

C2C12 cells were cultured on glass coverslips coated with L-Lysine (Nacalai Tesque). On day four of differentiation, myotubes after each treatment were fixed in 4% paraformaldehyde/ phosphate buffer solution (PBS). C2C12 myotubes were permeabilized in 0.25% Triton X-100/ PBS and blocked in 1% bovine serum albumin/PBS (blocking solution) for 1 h at room temperature. Myotubes were washed with PBS and incubated for 1 h in anti-muscle MHC antibody at a dilution of 1:250, washed with PBS, and incubated in Alexa Flour 488 conjugated goat anti-mouse antibody (Cell Signaling Technology, #4408) for 1 h, and washed with PBS. Antibodies were diluted in blocking solution. The samples were mounted in a 4', 6-diamidino-2-phenylindole (DAPI)-containing mounting medium. Images were captured with a BZ-9000 fluorescent microscope (Keyence, Osaka, Japan). Representative images for each treatment were selected. The myotube width was measured by BZ II Analyzer software (Keyence). One hundred myotubes per treatment were measured from ten non-overlapping fields-of-view, and were used to evaluate myotube atrophy.

## Statistical analysis

Data were expressed as mean ± SEM and analyzed by one-way analysis of variance followed by the Newman–Keuls test using Prism (version x9, Graphpad Inc., La Jolla, CA, USA). P-values < 0.05 were considered significant.

## Results

### Epi exacerbated LPS-induced myotube wasting and augmented LPS-induced Atrogin-1 mRNA expression

To examine whether catecholamines can affect LPS-induced myotube wasting, we exposed C2C12 myotubes to Epi and/or LPS for 24 h. In the absence of LPS, Epi (1 μM) did not significantly affect MHC protein expression. On the other hand, LPS (50 ng/mL) decreased MHC protein slightly, and addition of Epi (1 μM) further decreased MHC protein expression (Fig 1A). Immunofluorescence staining of the myotubes with an MHC-specific antibody revealed that LPS (50 ng/ml) slightly reduced the diameter of the myotubes, which was augmented by Epi (1 μM) (Fig 1B and 1C). Thus, Epi exacerbated LPS-induced C2C12 myotube wasting.

In view of the report that LPS induces C2C12 myotube wasting via activation of UPP [10], we analyzed the expression of Atrogin-1 and MuRF1, essential regulators of UPP in skeletal muscle, with qRT-PCR. Epi (1 μM) did not induce the Atrogin-1 mRNA expression, but augmented the LPS-induced Atrogin-1 mRNA expression in a concentration-dependent manner (Fig 2A and 2B). NE (1 μM) and ISO (1 μM), similarly to Epi, augmented LPS-induced Atrogin-1 mRNA expression (Fig 2C). Epi (1 μM) did not induce the MuRF1 mRNA expression in the absence of LPS, but increased MuRF1 mRNA levels in the presence of LPS (Fig 2D).

Two key downstream substrates of the mTOR complex 1 signaling, p70S6 kinase (S6K) and eukaryotic translation initiation factor 4E-binding protein (4E-BP), have been linked to protein synthesis in skeletal muscle [21]. To evaluate whether Epi and LPS affects protein synthesis, we analyzed p70S6K and 4E-BP phosphorylation in whole-cell lysates obtained from C2C12 myotubes exposed to Epi and/or LPS. Epi and/or LPS did not affect the

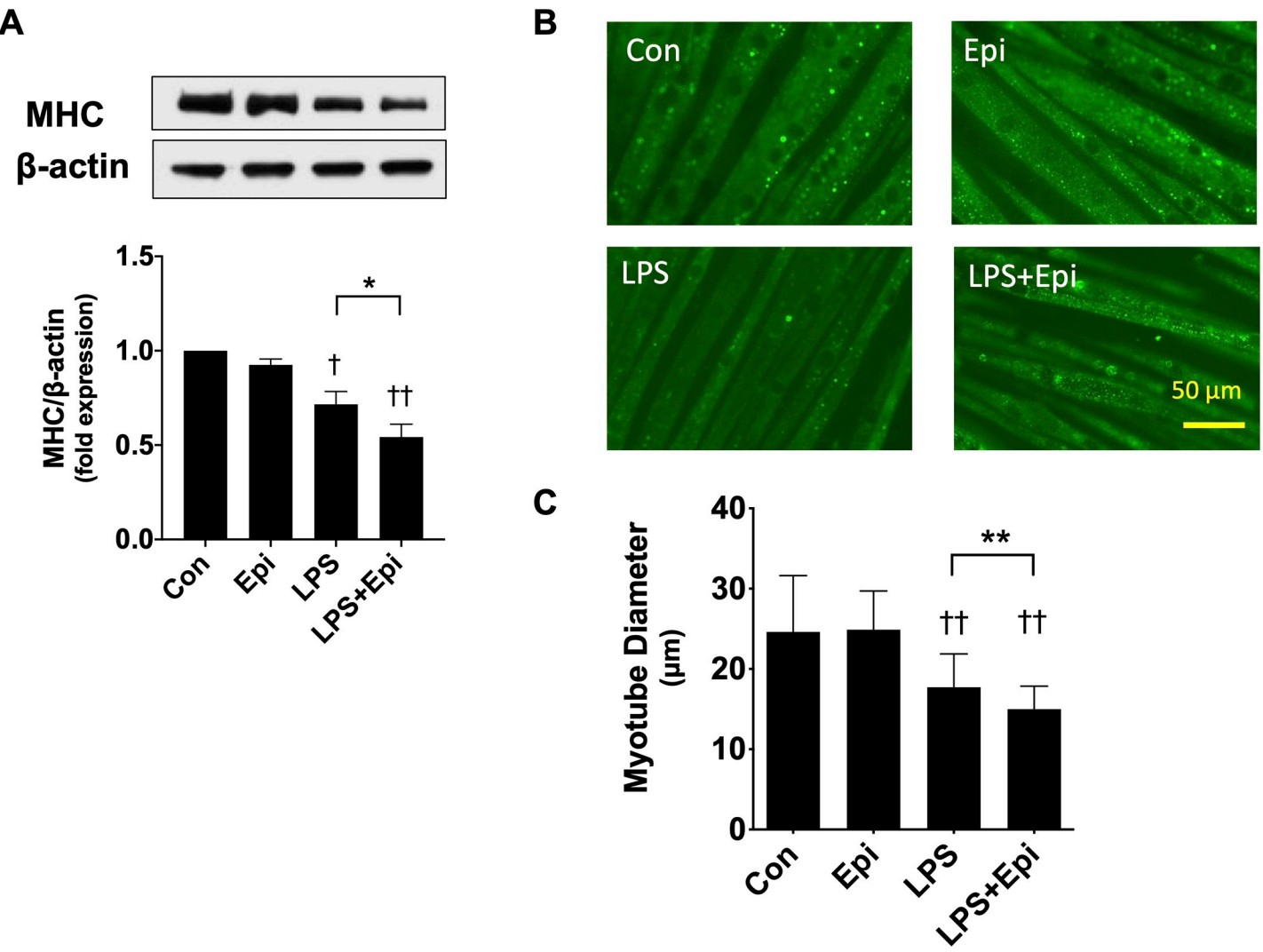

**Fig 1. Epinephrine (Epi) exacerbated lipopolysaccharide (LPS)-induced myotube wasting in C2C12 myotubes.** C2C12 myotubes were exposed to Epi (1 μM) and/or LPS (50 ng/ml) for 24 h. (A) Whole-cell lysates were analyzed for myosin heavy chain (MHC) and β-actin protein expression with immunoblotting assays. Representative immunoblots are shown (upper panel), and MHC protein expression quantified by densitometric analysis is shown as mean ± SEM (n = 5) of fold induction relative to the value of untreated control cells (lower panel). (B, C) Cells were immunostained with an MHC-specific antibody. Representative images (B), and myotube diameters measured from microscopic photographs (C) are shown. (†P < 0.05 and ††P < 0.005 compared with control, *P < 0.05 and **P < 0.005 for comparisons between the indicated groups).

phosphorylation of p70S6K and 4E-BP (S1 Fig). Thus, these results suggest that the effect of Epi on the LPS-induced muscle wasting contributes to proteolysis through the upregulation of UPP.

## STAT3 activation and C/EBPδ expression by LPS and Epi

STAT3 activated by phosphorylation on a tyrosine residue (Tyr 705) has been shown to be essential for muscle wasting in cachexia in cancer [13, 22]. Silva et al. clearly showed that C/EBPδ is involved in STAT3-dependent UPP upregulation in muscle wasting [13]. Therefore, to examine whether Epi affects STAT3 activation and C/EBPδ expression, we analyzed phospho-STAT3 (p-STAT3), total-STAT3 (t-STAT3), and C/EBPδ expression in whole-cell lysates obtained from C2C12 myotubes exposed to LPS and/or Epi. Immunoblotting assays showed

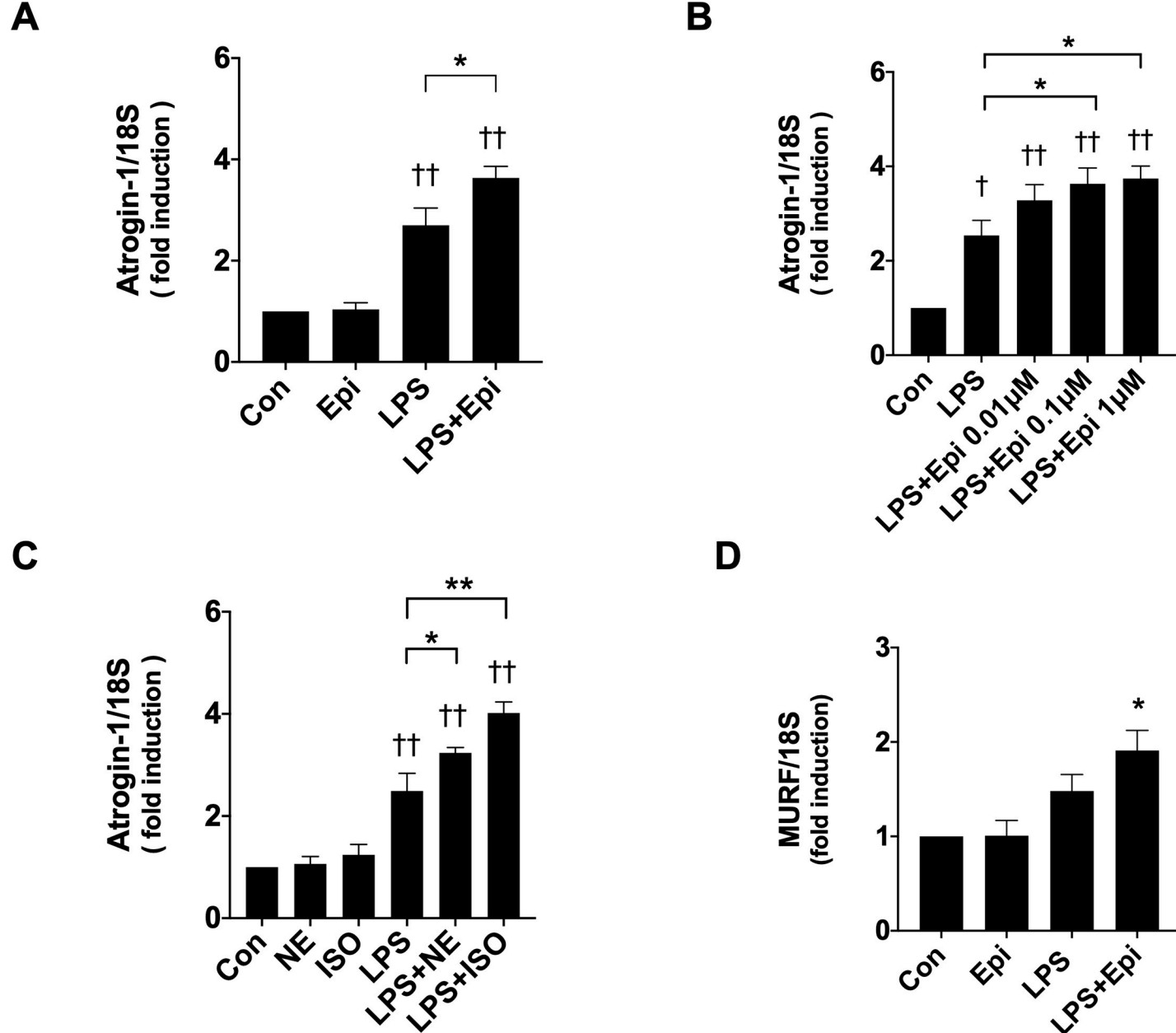

**Fig 2. Catecholamines increased LPS-induced Atrogin-1 messenger ribonucleic acid (mRNA) levels.** C2C12 myotubes were exposed for 3 h to LPS (50 ng/ml) and/ or Epi (1 μM) (A and D), Epi of the indicated concentration (B) and NE (1 μM) or ISO (1 μM) (D). Atrogin-1 (A, B and C) and MuRF1 (D) mRNA expression levels were analyzed by quantitative reverse transcription polymerase chain reaction (qRT-PCR) (n = 4). Data are normalized to the 18S rRNA expression level, and fold induction relative to the value of the control condition is presented as mean ± SEM (n = 4). (†P < 0.05 and ††P < 0.005 compared with control, *P < 0.05 and **P < 0.005 for comparisons between the indicated groups).

that Epi (1 μM) and LPS (50 ng/ml) increased p-STAT3 (Fig 3A). Epi increased LPS-induced C/EBPδ expression, although C/EBPδ was not induced by Epi in the absence of LPS (Fig 3B). These results suggest that the augmenting effect of Epi on the LPS-induced myotube wasting may be mediated by UPP upregulation through STAT3 and C/EBPδ.

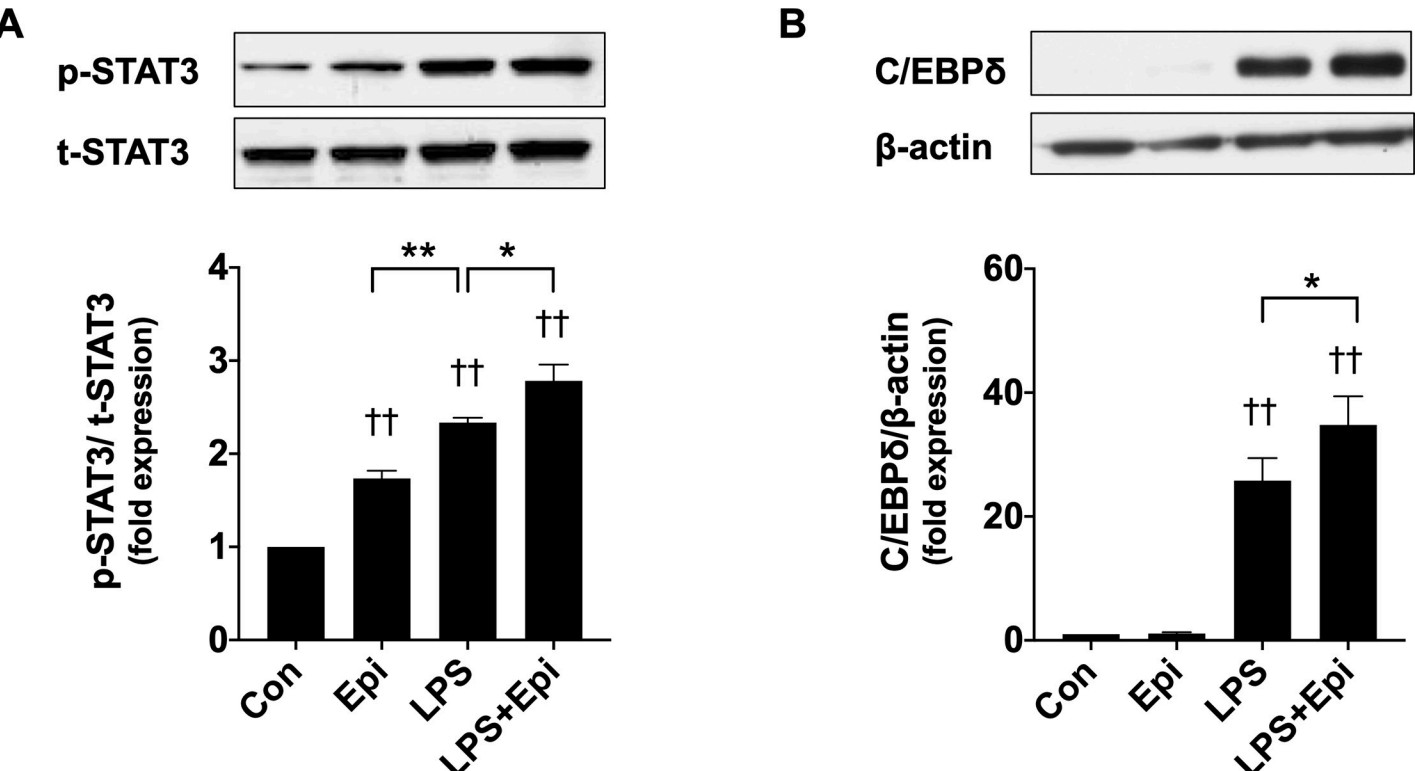

**Fig 3. Activation of the STAT3-C/EBPδ pathway by LPS and Epi.** C2C12 myotubes were exposed to Epi (1 μM) and/or LPS (50 ng/ml) for 3 h. Whole-cell lysates were analyzed for p-STAT3, t-STAT3, C/EBPδ, and β-actin protein expressions with immunoblotting analysis. Representative immunoblots are shown. Expressions of p-STAT3 (A) and C/EBPδ (B) proteins were quantified by densitometric analysis, and were normalized to the expression level of t-STAT3 and β-actin, respectively. Fold induction relative to the value of the control condition is presented as mean ± SEM (n = 5). (†P < 0.05 and ††P < 0.005 compared with control, *P < 0.05 and **P < 0.005 for comparisons between the indicated groups).

### Involvement of IL-6 in STAT3 activation and C/EBPδ expression by LPS and Epi

Because IL-6 can be produced and released by skeletal muscle cells in response to various stresses [15], and can activate the canonical STAT3 signaling pathway [23], we hypothesized that myotube-derived IL-6 was involved in the myotube wasting process induced by LPS and Epi, and was mediated by the STAT3 signaling pathway. First, we examined whether Epi and LPS increased cytokine production in C2C12 myotubes. It was revealed that LPS (50 ng/ml) increased both IL-6 and TNF-α mRNA levels (Fig 4A and 4B). In the absence of LPS, Epi (1 μM) significantly increased IL-6 mRNA levels (2.7 ± 0.4 fold) (Fig 4A), but did not increase TNF-α mRNA levels (Fig 4B). In the presence of LPS, Epi remarkably increased IL-6 mRNA level in a concentration-dependent manner (Fig 4A and 4C), but decreased TNF-α mRNA levels (Fig 4B). Other catecholamines, namely, ISO (1 μM), and NE (1 μM), slightly induced IL-6 mRNA levels in the absence of LPS (NE, 1.9 ± 0.5 fold; ISO, 2.3 ± 0.4 fold), but remarkably increased LPS-induced IL-6 mRNA levels (Fig 4D). ELISA demonstrated that Epi (1 μM) significantly increased LPS-induced IL-6 protein secretion (Fig 4E). These results indicated that catecholamines and LPS synergized to increase IL-6 production and secretion. To examine whether IL-6 produced by Epi and LPS was involved in myotube wasting, we tested the effect of the IL-6 inhibitor LMT-28 [24]. Pretreatment of the myotube with LMT-28 (50 μM) remarkably suppressed Epi-induced STAT3 activation (Fig 5A and 5B), and Epi-induced expression of C/EBPδ and Atrogin-1 in the presence of LPS (Fig 5A, 5C and 5D), whereas

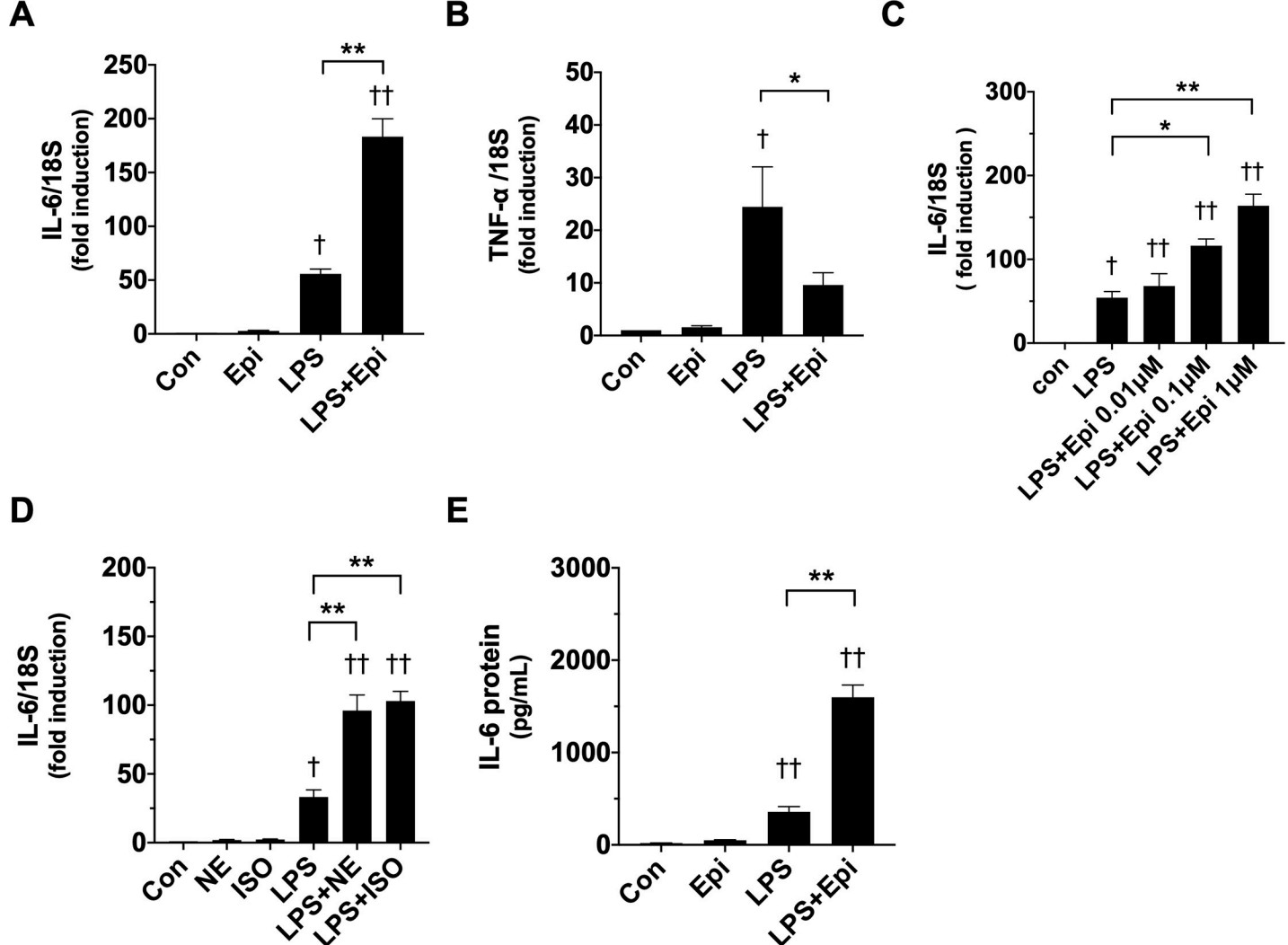

**Fig 4. Catecholamines potentiated LPS-induced interleukin (IL)-6 production.** C2C12 myotubes were exposed for 3 h to LPS (50 ng/ml) and/or Epi (1 μM) (A, B, and E), Epi of the indicated concentration (C) and NE (1 μM) or ISO (1 μM) (D). The mRNA expression levels of IL-6 (A, C, and D) and TNF-α (B), were analyzed by qRT-PCR. Data are normalized to the 18S rRNA expression level and fold induction relative to the value of the control condition is presented as mean ± SEM (n = 3). IL-6 concentration in the conditioned media was analyzed by the enzyme-linked immunosorbent assay (ELISA) assay (E). Data are presented as mean ± SEM (n = 4–6). (†P < 0.05 and ††P < 0.005 compared with control, *P < 0.05 and **P < 0.005 for comparisons between the indicated groups).

LMT-28 partially suppressed the effect of LPS in the absence of Epi. These results suggest that the myotube-derived IL-6 produced by stimulation of LPS and Epi affects the myotubes in an autocrine or paracrine manner and induces myotube wasting.

## NF-κB activation is necessary for myotube wasting induced by LPS and Epi

To further examine the involvement of IL-6 in myotube wasting via the STAT3-C/EBPδ-Atro-gin-1 pathway, we tested the effects of mouse recombinant IL-6 (rIL-6). In the absence of LPS, rIL-6 (10 ng/ml) induced p-STAT3 (Fig 6A), but C/EBPδ protein and Atrogin-1 mRNA levels were not significantly changed by rIL6 at concentrations as high as 100 ng/ml (Fig 6A and 6B). In the presence of LPS (50 ng/ml), rIL-6 (10 ng/ml) remarkably increased p-STAT3 and C/EBPδ expression, and led to an increase of the LPS-induced Atrogin-1 mRNA level (Fig 6A

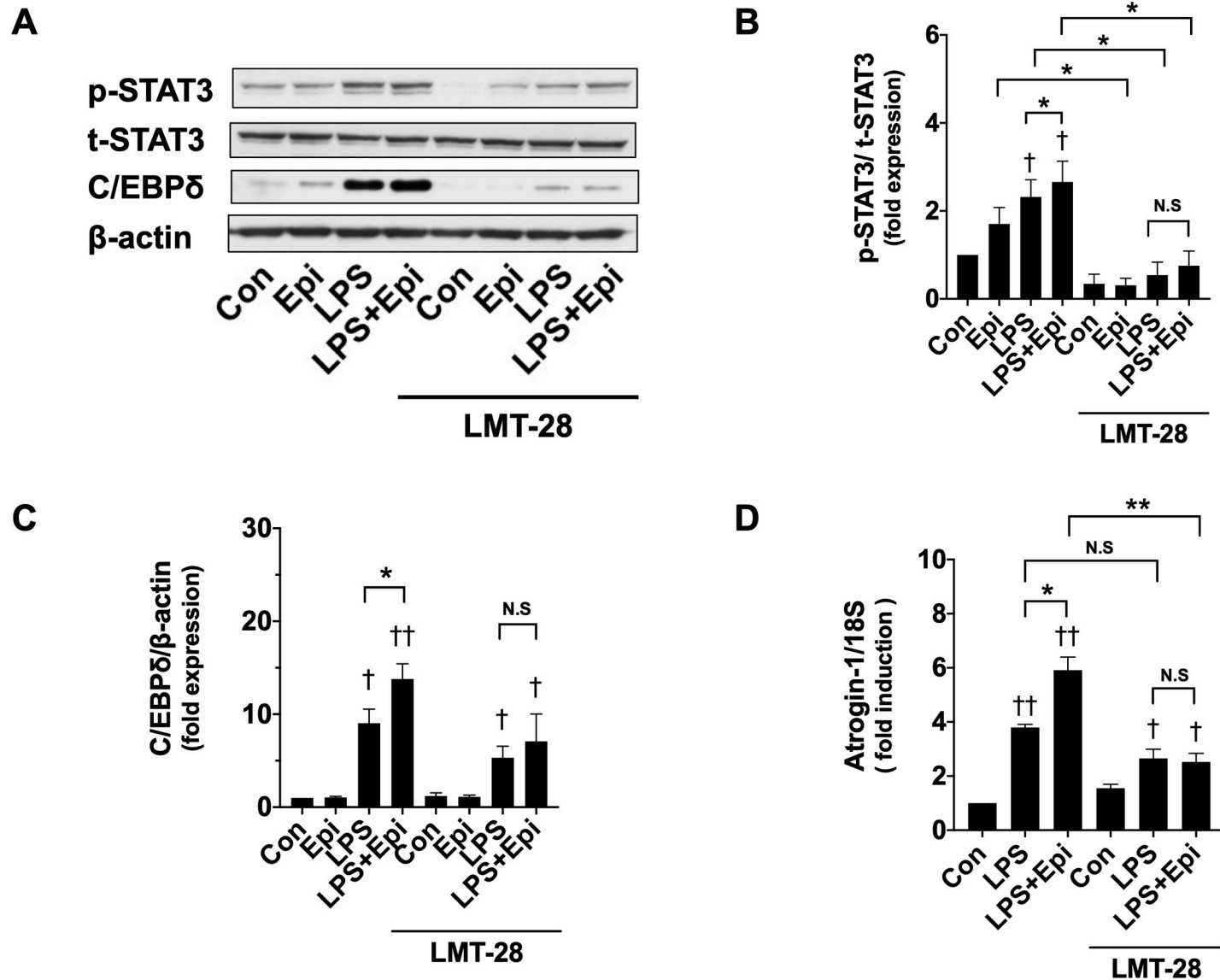

**Fig 5. Effects of LMT-28 on the responses induced by Epi and LPS.** C2C12 myotubes were pretreated with the IL-6 inhibitor LMT-28 (50 μM) for 30 min, and were exposed to Epi (1 μM) and/or LPS (50 ng/ml) for 3 h. Whole-cell lysates were analyzed for p-STAT3, t-STAT3, and C/EBPδ with the immunoblotting assay (A, B, and C). Representative immunoblots are shown (A), and the expressions of p-STAT3 (B) and C/EBPδ protein (C) were quantified by densitometric analysis and normalized to the expression levels of t-STAT3 and β-actin, respectively. Fold induction relative to the value of the control condition is presented as mean ± SEM (n = 5). Atrogin-1 mRNA expression levels were analyzed by qRT-PCR (D). Data are normalized to the 18S rRNA expression levels, and fold induction relative to the value of the control condition is presented as mean ± SEM (n = 4). (†P < 0.05 and ††P < 0.005 compared with control, *P < 0.05 and **P < 0.005 for comparisons between the indicated groups).

and 6B). These results suggest that Atrogin-1 induction by LPS cannot be explained by the IL-6-STAT3 signaling pathway alone, and that another mechanism should be involved.

LPS is known to induce the activity of NF-κB activity that constitutes an essential transcription factor in the development of muscle wasting [25]. Thus, we aimed to evaluate the possible involvement of NF-κB in the C2C12 myotube wasting. First, to test whether Epi and rIL-6 affect the activity of NF-κB, we performed an ELISA assay to a NF-κB (p65) DNA-binding activity. Although LPS (50 ng/ml) increased the activity of NF-κB, Epi (1 μM) or rIL-6 (10 ng/ml) did not affect the activity of NF-κB in the presence or absence of LPS (Fig 6C), thus

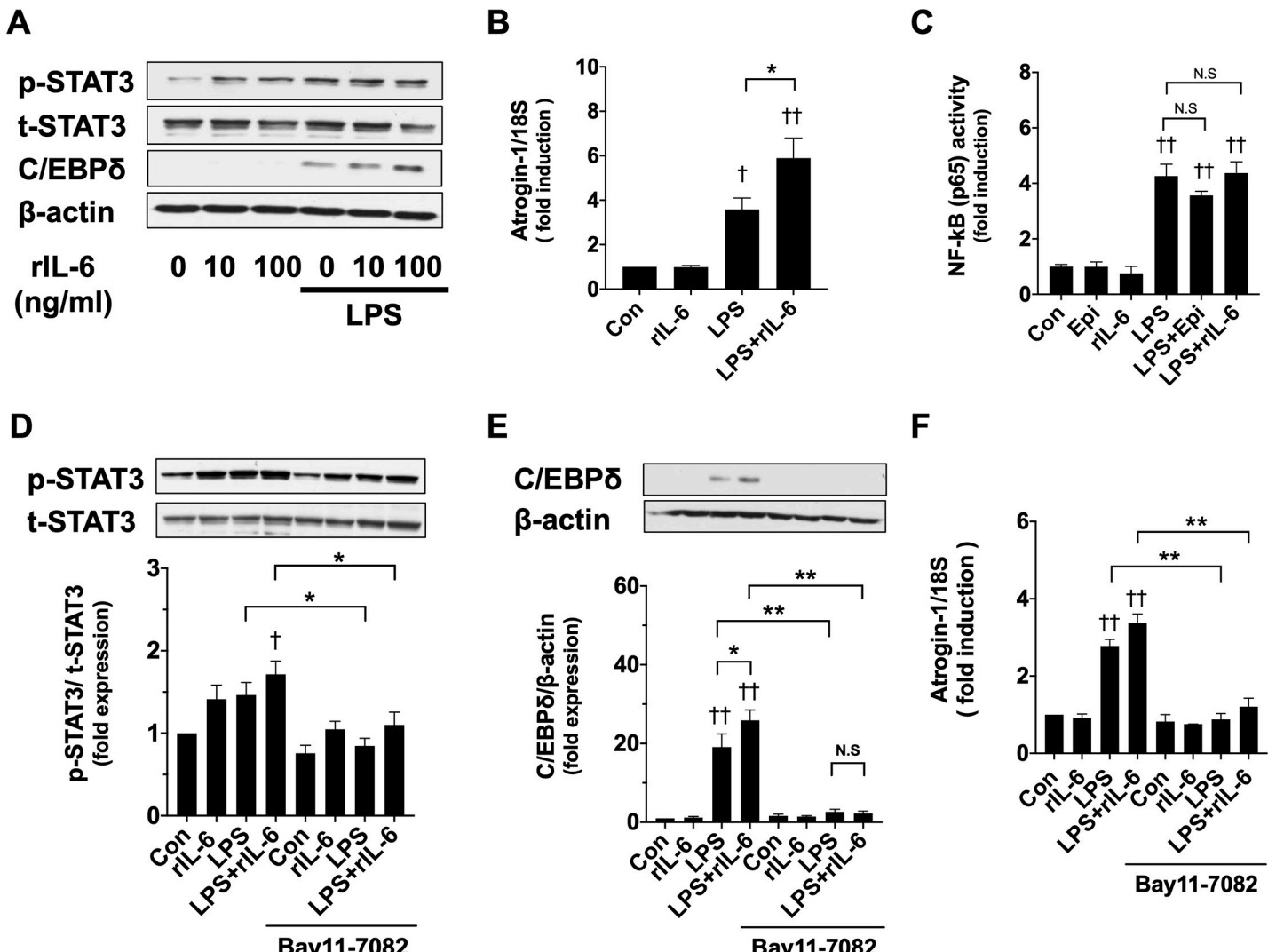

**Fig 6. NF-κB activation is involved in myotube wasting induced by LPS and recombinant IL (rIL)-6.** (A) C2C12 myotubes were exposed to rIL-6 (10 and 100 ng/ml) and/or LPS (50 ng/ml) for 3 h. Whole-cell lysates were analyzed for p-STAT3, t-STAT3, C/EBPδ, and β-actin protein expressions by immunoblotting analysis. Representative immunoblots are shown. (B) C2C12 myotubes were exposed to rIL-6 (100 ng/ml) and/or LPS (50 ng/ml) for 3 h, and the Atrogin-1 mRNA levels were analyzed by qRT-PCR. Data are normalized to the 18S ribosomal RNA (rRNA) expression levels, and fold induction relative to the value of the control condition is presented (n = 4). (C) C2C12 myotubes were treated by Epi (1 μM) and/or LPS (50 ng/ml) for 3 h, and NF-κB (p65) binding activity was analyzed by a TranAM ELISA kit. Fold induction was calculated relative to the value of the untreated control cells (n = 5). (D, E) C2C12 myotubes, pretreated with the NF-κB inhibitor Bay 11–7082 (10 μM) for 30 min were exposed to rIL-6 (100 ng/ml) and/or LPS (50 ng/ml) for 3 h. Representative immunoblots for p-STAT3 (D) and C/EBPδ (E) protein are shown, and the expressions of p-STAT3 (D) and C/EBPδ protein (E) were quantified by densitometric analysis and normalized to the expression levels of t-STAT3 and β-actin, respectively. Fold induction relative to the value of the control condition is presented as mean ± SEM (n = 6). (F) Atrogin-1 mRNA was analyzed by qRT-PCR, and its expression level normalized to the 18S rRNA expression level is presented as mean ± SEM (n = 4) of fold induction relative to the value of the control condition. (†P < 0.05 and ††P < 0.005 compared with control, *P < 0.05 and **P < 0.005 for comparisons between the indicated groups).

indicating that Epi or rIL-6 could not directly induce NF-κB activation. To confirm the involvement of NF-κB activity in myotube wasting, the effect of the NF-κB inhibitor Bay 11–7082 was then tested. Bay 11–7082 (10 μM) slightly suppressed STAT3 activation induced by LPS (Fig 6D). This result suggests that NF-κB signaling might be involved in the LPS-induced STAT3 activation. On the other hand, Bay 11–7082 (10 μM) abolished not only the C/EBPδ protein, but also Atrogin-1 mRNA expression induced by LPS with or without rIL-6 (Fig 6E

and 6F). Taken together, these results suggest that the myotube wasting induced by LPS and Epi requires activation of the NF-κB signaling pathway.

## Epi might potentiate LPS-induced IL-6 synthesis via cyclic adenosine monophosphate (cAMP) signaling

Given that NF-κB activated by LPS has been shown to be one of the main regulators of the IL-6 gene [26], we examined the effects of the NF-κB inhibitor on IL-6 synthesis. Bay 11–7082 suppressed remarkably LPS-induced IL-6 mRNA and protein expression (Fig 7A and 7B), but Epi (1 μM) augmented significantly LPS-induced IL-6 synthesis in the presence of Bay 11–7082, thus suggesting that Epi induces IL-6 production by a mechanism other than NF-κB activation. Because the activation of β adrenergic receptors induces adenylyl cyclase activation via the stimulatory G-protein, and thus results in an increase in intracellular cAMP [1], we examined whether Forskolin that directly activates adenylyl cyclase and raises intracellular cAMP can affect IL-6 synthesis. Forskolin (10 μM) slightly increased IL-6 mRNA levels and remarkably increases LPS-induced IL-6 mRNA levels (Fig 7C) in a manner similar to Epi (1 μM).

Furthermore, we evaluated the phosphorylation of cAMP response element binding protein (CREB), a downstream effector of cAMP signaling. Not only Epi but also LPS activates CREB. These results suggest that cAMP signaling, but not NF-κB signaling, mediates synergistic IL-6 expression.

## A β-blocker suppressed the potentiating effect of Epi on LPS-induced muscle wasting

The adrenergic receptor expressed on skeletal muscle predominantly belongs to the β-adrenergic receptor. To examine whether Epi affects LPS-induced myotube wasting via the β-adrenergic receptor, we tested the effect of pretreatment of C2C12 myotubes with the nonselective β-blocker Carvedilol for 30 min. As shown in Fig 8A and 8B, pretreatment of Carvedilol (5 μM) did not affect the effect of LPS, but suppressed the potentiating effect of Epi (1 μM) on LPS-induced IL-6 mRNA and protein induction. Carvedilol also suppressed the effect of Epi (1 μM) on the Atrogin-1 mRNA induction by LPS (50 ng/ml) (Fig 8C), and abolished the effect of Epi (1 μM) on the MHC protein loss in the presence of LPS (50 ng/ml) (Fig 8D and 8E). These results indicate that Epi exacerbates the LPS-induced muscle wasting through the β adrenergic receptor in C2C12 myotubes.

## Discussion

In the present study, we demonstrated that catecholamine exacerbated LPS-induced muscle wasting in C2C12 myotubes. Fig 9 summarizes the proposed mechanism for muscle wasting induced by LPS and catecholamines. LPS can activate NF-κB and induce the expressions of C/EBPδ and Atrogin-1, and LPS-induced STAT3 activation through the NF-κB-mediated IL-6 production can facilitate their expressions. Catecholamine drastically increases LPS-induced IL-6 production, and leads to further activation of STAT3 and to the augmentation of the expressions of C/EBPδ and Atroign-1.

Muscle wasting in sepsis is primarily the result of increased protein breakdown through UPP [8]. Because ubiquitin ligases Atrogin-1 and MuRF1 are induced early during the wasting process [11] and regulate UPP in skeletal muscle, we analyzed the expression of Atrogin-1 for the assessment of muscle wasting. The present study demonstrated that catecholamines increased LPS-induced Atrogin-1 expression, and Epi also slightly increased LPS-induced

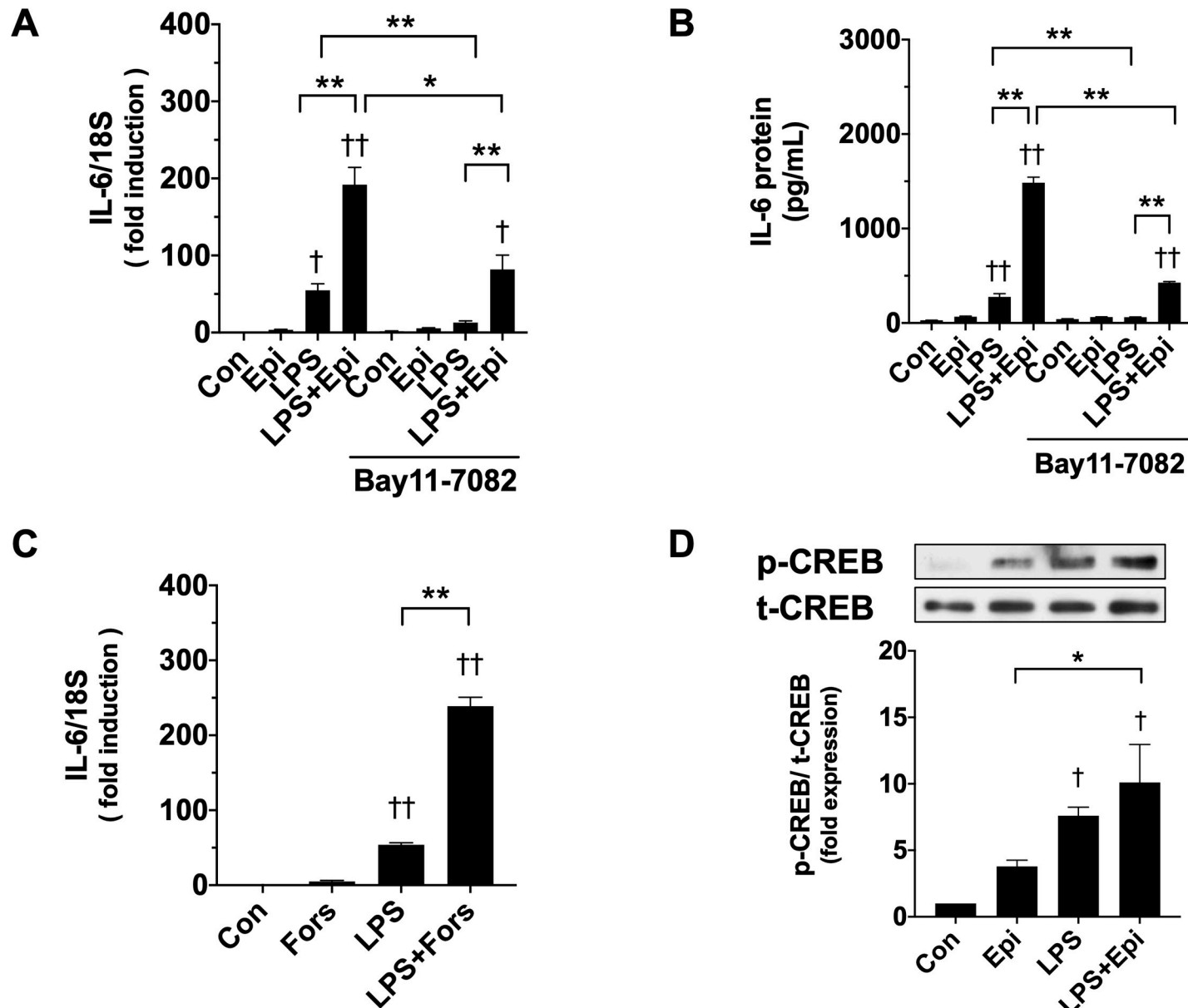

**Fig 7. Involvement of NF-κB activation and cyclic adenosine monophosphate (cAMP) signaling in the IL-6 production induced by LPS and Epi.** (A, B) After pretreatment with the NF-κB inhibitor Bay 11–7082 (10 μM) or a vehicle for 30 min, C2C12 myotubes were exposed to Epi (1 μM) and/or LPS (50 ng/ml). IL-6 mRNA was analyzed by qRT-PCR, and its expression levels were normalized to the 18S rRNA expression levels and presented as mean ± SEM (n = 3) of fold induction relative to the value of the control condition (A). The IL-6 concentration in the conditioned media was analyzed with the ELISA assay, and is presented as mean ± SEM (n = 3–4) (B). (C) C2C12 myotubes were treated with Forskolin (10 μM) and/or LPS (50 ng/ml) for 3 h. IL-6 mRNA expression was analyzed by qRT-PCR, and its expression levels normalized to the 18S rRNA expression levels are presented as mean ± SEM (n = 3) of fold induction relative to the value of the control condition. (D) C2C12 myotubes were exposed to Epi (1 μM) and/or LPS (50 ng/ml) for 3 h. Whole-cell lysates were analyzed for p-CREB, t-CREB protein expressions with immunoblotting analysis. Representative immunoblots are shown. Expressions of p-CREB proteins were quantified by densitometric analysis, and were normalized to the expression level of t-CREB. Fold induction relative to the value of the control condition is presented as mean ± SEM (n = 3). (†P < 0.05 and ††P < 0.005 compared with control, *P < 0.05 and **P < 0.005 for comparisons between the indicated groups).

MuRF1 expression. This suggests that catecholamines can facilitate muscle wasting in septic patients through UPP activation. It was previously reported that STAT3 signaling was involved in skeletal muscle wasting in cancer cachexia and sepsis [12, 22, 27], and recent studies showed that STAT3 activation causes C/EBPδ to promote muscle wasting [12, 13, 28]. Silva et al.

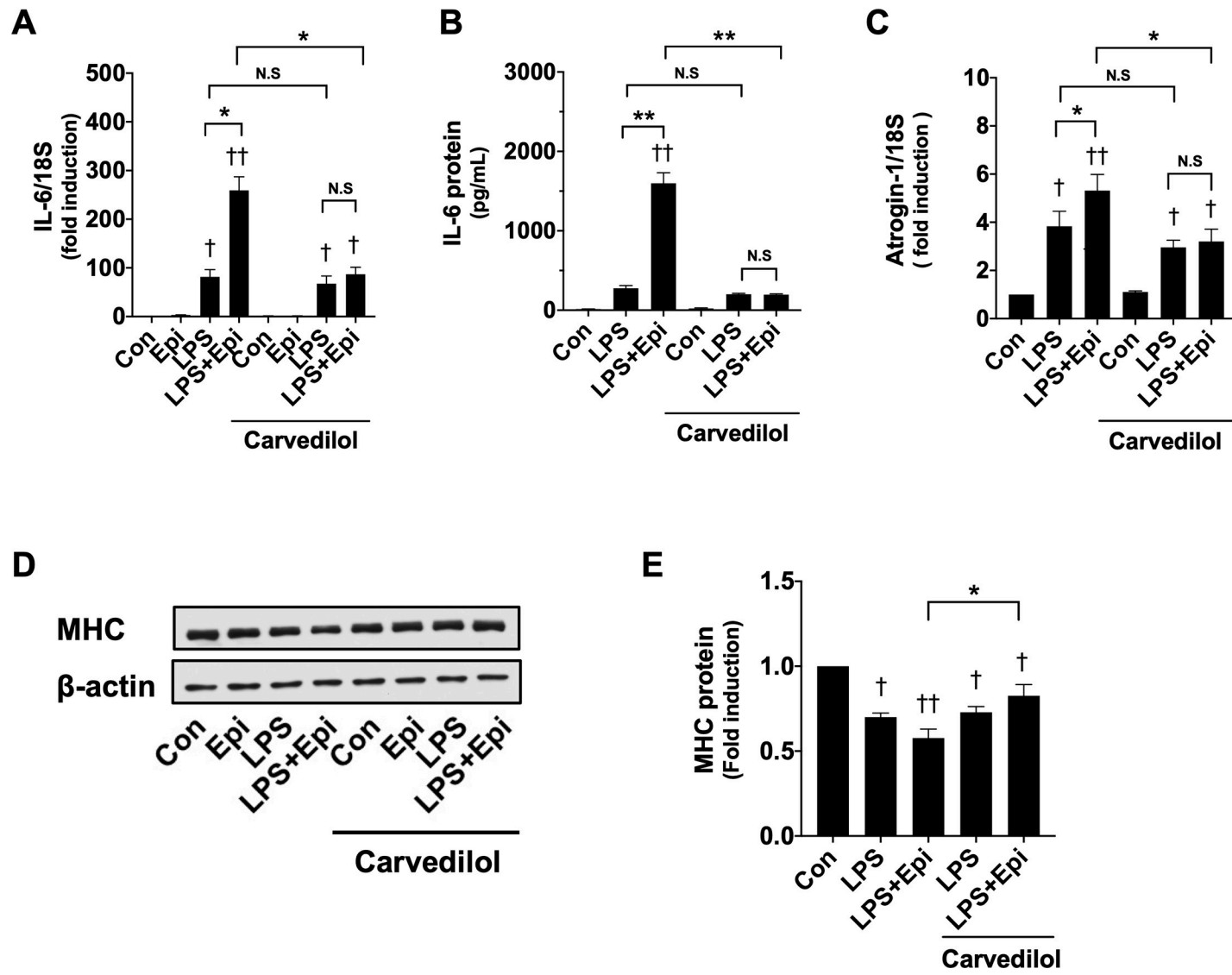

**Fig 8. A β-blocker inhibited the effect of Epi in C2C12 myotubes.** (A, B, C) C2C12 myotubes were pretreated with the β-blocker Carvedilol (5 μM) for 30 min, and were then exposed to Epi (1 μM) and/or LPS (50 ng/ml) for 3 h. IL-6 (A) and Atrogin-1 (C) mRNA were analyzed with qRT-PCR, and their expression levels, normalized to the 18S rRNA expression level, are presented as mean ± SEM (n = 3) of fold induction relative to the value of the control condition. The IL-6 concentrations in the conditioned media were analyzed with the ELISA assay, and are presented as mean ± SEM (n = 3–4) (B). (D, E) After pretreatment with Carvedilol (5 μM) for 30 min, C2C12 myotubes were exposed to Epi (1 μM) and/or LPS (50 ng/ml) for 24 h. Whole-cell lysates were analyzed for MHC and β-actin protein expression by immunoblotting assays, and a representative immunoblot is shown (D). MHC protein expressions, quantified by densitometric analysis, were normalized to the expression level of β-actins, and fold induction relative to the control condition is presented as mean ± SEM (n = 5) (E). (†P < 0.05 and ††P < 0.005 compared with control, *P < 0.05 and **P < 0.005 for comparisons between the indicated groups).

demonstrated that C/EBPδ binding sites exist in the Atrogin-1 promoter, and overexpression of C/EBPδ in C2C12 cells can increase the promoter activity of the Atrogin-1 gene [13]. The finding in the present study that Epi increased the LPS-induced p-STAT3, C/EBPδ, and Atrogin-1 expressions, in conjunction with findings from previous reports, suggests that catecholamine can activate STAT3, thus leading to myotube wasting.

As a mechanism for catecholamine-induced STAT3 activation, we focused on the involvement of IL-6 signaling, a representative activation mechanism of the STAT3 pathway. Several studies reported that IL-6 promoted skeletal muscle wasting in clinical and experimental

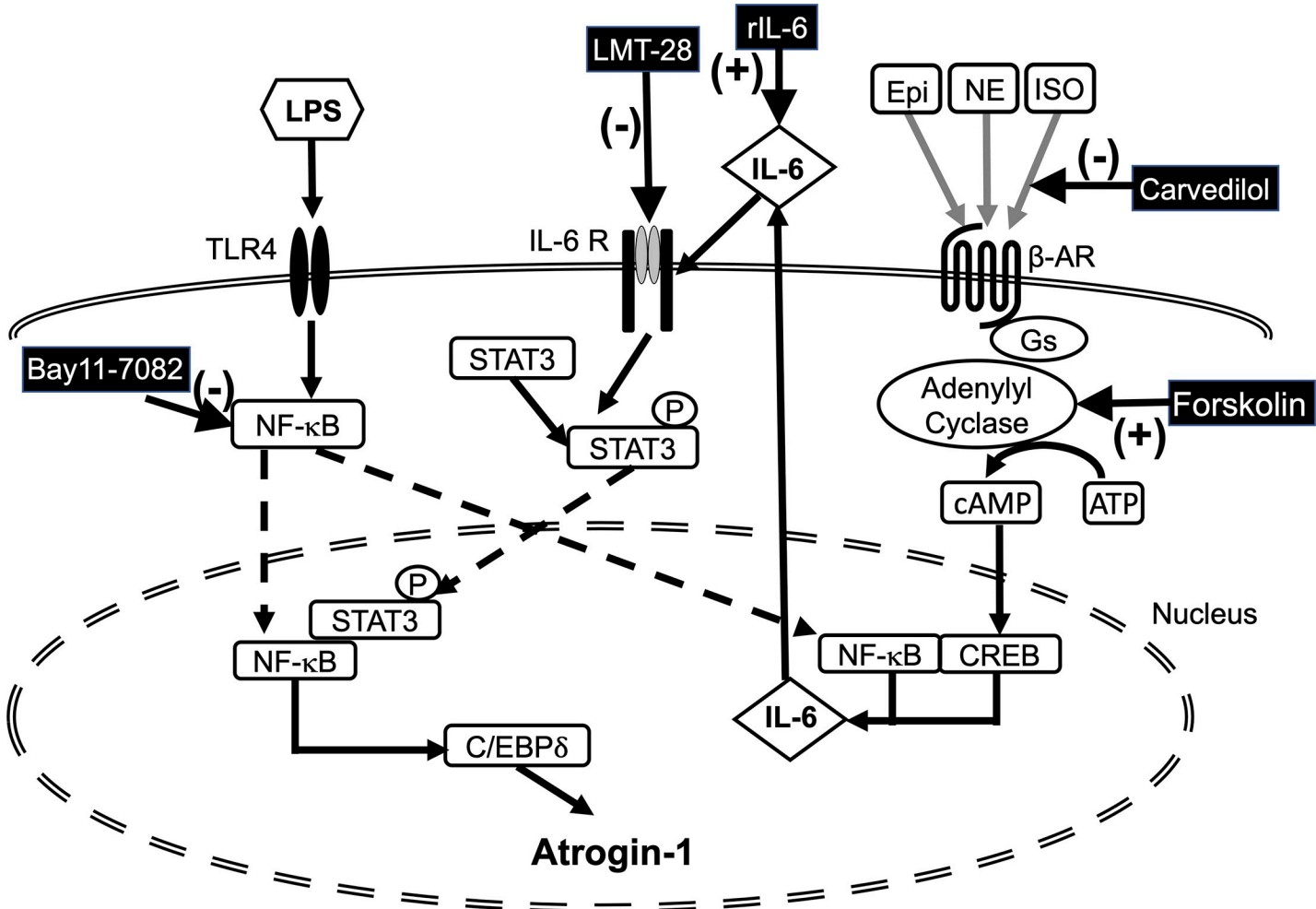

**Fig 9. Schematic representation of the effects of catecholamines on LPS-induced myotube wasting.** NF-κB activated by LPS via toll-like receptor 4 (TLR4) induces the expressions of C/EBPδ and Atrogin-1. NF-κB also induces IL-6 production and STAT3 activation that further facilitates the induction of C/EBPδ and Atroign-1, thus leading to muscle wasting. The β-adrenergic receptor (βAR) activated by catecholamines increases cAMP production, and drastically augments LPS-induced IL-6 production, thus leading to further activation of STAT3 and potentiation of the expressions of C/EBPδ and Atrogin-1. This results in the exacerbation of muscle wasting. Reagents used in this study, LMT-28, Bay11-7082, Carvedilol, Forskolin, and rIL-6, are also shown in the scheme. The symbols (+) and (-) are used to indicate that the reagent promotes or inhibits the indicated mediator, respectively.

cachexia [29–31], although it was also reported that low levels of IL-6 could promote activation of satellite cells and myotube regeneration [32, 33]. LPS induces systemic inflammation that leads to increased circulating proinflammatory cytokines, including TNF-α and IL-6 [18, 31, 34]. However, LPS can also directly affect skeletal muscle to produce cytokines [35]. Given that myotubes were not administered with exogenous proinflammatory cytokines in the present study, we hypothesized that myotube-derived cytokines were involved in the catecholamine-induced STAT3 activation. We demonstrated that catecholamines drastically increased LPS-induced IL-6 mRNA expression and IL-6 secretion by the myotubes. To examine whether the myotube-derived IL-6 was involved in the effects induced by Epi, we used the IL-6 inhibitor LMT-28 that functions through direct binding to gp130, a standard hub for transducing signals for IL-6 family cytokines [24]. It was shown that LMT-28 abolished the potentiating effect of Epi, thus suggesting that myotube-derived IL-6 was involved in Epi-induced myotube wasting. We also showed that rIL-6 increased LPS-induced STAT3 phosphorylation, and C/EBPδ

and Atrogin-1 gene expression, thus supporting the involvement of IL-6 signaling in Epi-induced muscle wasting. On the other hand, LMT-28 only partially abolished LPS-induced STAT3 activation. Bonetto et al. also reported that administration of LPS in IL-6 knockout mice leads to STAT3 activation [22]. Thus, IL-6 can activate STAT3 by the IL-6 pathway and the other mechanism.

In the presence of LPS, rIL-6 as well as Epi induced Atrogin-1 and C/EBPδ expressions, whereas in the absence of LPS, rIL-6 induced STAT3 phosphorylation, but did not induce Atrogin-1 or C/EBPδ expressions. Therefore, as a mechanism other than IL-6 signaling necessary for Epi-induced muscle wasting, we then focused on NF-κB, which was activated by LPS and was reported to be one of the essential transcription factors in the development of muscle wasting [25]. It has also been reported that NF-κB activates the expression of the ubiquitin ligase and C/EBP protein family including C/EBPδ [25, 36, 37]. In the present study, Epi or rIL-6 did not activate NF-κB in the presence or absence of LPS, but the NF-κB inhibitor Bay 11–7082 abolished C/EBPδ and Atrogin-1 induction by LPS with or without Epi. This suggested that NF-κB activation was an essential factor for myotube wasting caused by LPS and Epi. Several studies have proven that the STAT3 signaling can collaborate with NF-κB to promote pro-cachectic or inflammatory genes [38, 39]. Thus, although further work will need to examine how the STAT3 and NF-κB cooperate to promote the expression of pro-cachectic genes, we can conclude that the exacerbating effects of catecholamines on muscle wasting are limited to the inflammatory condition, such as sepsis, in which NF-κB is activated in skeletal muscle.

NF-κB is one of the main regulators of the IL-6 gene [40]. However, we demonstrated that in contrast to LPS-induced IL-6 synthesis, Epi-induced IL-6 synthesis was not suppressed by Bay 11–7082. This suggested that Epi induced IL-6 synthesis through a mechanism other than NF-κB. IL-6 gene regulation in muscle was inherently organized to respond to a wide variety of signals [15, 26]. A previous report demonstrated that the combined stimulation of NF-κB and the cAMP response element binding protein (CREB) synergistically induced transcription at the IL-6 promoter in human astrocytes [41]. Song et al. also reported that LPS triggered cAMP signaling in addition to NF-kB signaling, leading to the production of IL-6 [42]. Activation of the β adrenergic receptor leads to adenylyl cyclase activation via the stimulatory G-protein, and increases intracellular cAMP [1]. We demonstrated that Forskolin increased IL-6 mRNA levels, and Forskolin and LPS synergized to increase the IL-6 levels. This indicated that Forskolin can emulate the effects of Epi on IL-6 production. Furthermore, we also showed that Epi and LPS phosphorylated the CREB, which is activated by cAMP signaling. Therefore, the IL-6 synthesis synergistically induced by LPS and Epi may be mediated by the NF-κB signaling pathway and cAMP signaling, respectively. In contrast, another proinflammatory cytokine, TNF-α, was induced by LPS in a manner similar to that of IL-6, but was not induced by cAMP.

A significant proportion of the adrenergic receptors expressed in skeletal muscle belongs to the β-adrenergic receptor [1]. The suppressive effect of the nonselective β blocker Carvedilol on the potentiation of the LPS-induced muscle wasting by Epi, indicates that Epi exacerbates LPS-induced myotube wasting via the β adrenergic receptor. Therefore, to prevent exacerbation of skeletal muscle wasting by catecholamines in the presence of systematic inflammation, we should try to avoid activation of the β adrenergic receptor expressed in the skeletal muscle. In the clinical settings, it might be better to avoid the use of vasopressors with a strong β agonistic effect or coadminister catecholamines and β blockers in septic patients.

In conclusion, Epi exacerbates LPS-induced myotube wasting in C2C12 myotubes by potentiating the NF-κB-mediated activation of the C/EBPδ-Atrogin-1 pathway via the IL-6-STAT3 pathway. Further understanding of the molecular mechanism of skeletal muscle

wasting is necessary to develop strategies for preventing muscle wasting and improving prognosis in critically ill patients.

## Supporting information

**S1 Fig. Epi did not affect the phosphorylation of p70S6K and 4E-BP in the presence of LPS.** C2C12 myotubes were exposed to Epi (1 μM) and/or LPS (50 ng/ml) for 3 h. Whole-cell lysates were analyzed for p-p70S6K, t-p70S6K, p-4EBP, and t-4EBP protein expressions with immunoblotting analysis. Representative immunoblots are shown. Expressions of p-p70S6K (A) and p-4EBP (B) proteins were quantified by densitometric analysis, and were normalized to the expression level of t-p70S6K and t-4EBP, respectively. Fold induction relative to the value of the control condition is presented as mean ± SEM (n = 3). (†$P < 0.05$ and ††$P < 0.005$ compared with control, *$P < 0.05$ and **$P < 0.005$ for comparisons between the indicated groups)
(TIFF)

**S1 File. The raw images of all Western blot data.**
(PDF)

**S1 Dataset.**
(XLSX)

## Acknowledgments

The authors would like to thank Yuko Ono, M.D., Ph.D. (Department of Disaster and Emergency Medicine, Graduate School of Medicine, Kobe University), and Kazuho Sakamoto, Ph. D. (Department of Bio-Informational Pharmacology, School of Pharmaceutical Sciences, University of Shizuoka, Shizuoka, Japan), for technical advice.

## Author Contributions

**Conceptualization:** Shinichi Kai, Kazuhiko Fukuda.

**Data curation:** Shino Matsukawa, Shinichi Kai.

**Formal analysis:** Shino Matsukawa.

**Funding acquisition:** Shinichi Kai, Hideya Seo.

**Investigation:** Shino Matsukawa, Shinichi Kai, Hideya Seo, Kengo Suzuki.

**Project administration:** Shinichi Kai, Kazuhiko Fukuda.

**Supervision:** Shinichi Kai, Kazuhiko Fukuda.

**Writing – original draft:** Shino Matsukawa, Shinichi Kai.

**Writing – review & editing:** Shinichi Kai, Kazuhiko Fukuda.

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
