## [Decision Letter · Decision Letter 0]

5 Mar 2021

PONE-D-21-02018

Activation of the β-adrenergic receptor exacerbates lipopolysaccharide-induced wasting of skeletal muscle cells by increasing interleukin-6 production

PLOS ONE

Dear Dr. Kai,

Thank you for submitting your manuscript to PLOS ONE. After careful consideration, we feel that it has merit but does not fully meet PLOS ONE’s publication criteria as it currently stands. Therefore, we invite you to submit a revised version of the manuscript that addresses the points raised during the review process.

We look forward to receiving your revised manuscript.

Kind regards,

Atsushi Asakura, Ph.D

Academic Editor

PLOS ONE

Journal Requirements:

<h1>** **</h1>

Reviewers' comments:

Reviewer's Responses to Questions

**Comments to the Author**

1. Is the manuscript technically sound, and do the data support the conclusions?

Reviewer #1: Partly

Reviewer #2: No

2. Has the statistical analysis been performed appropriately and rigorously? 

Reviewer #1: Yes

Reviewer #2: Yes

3. Have the authors made all data underlying the findings in their manuscript fully available?

Reviewer #1: Yes

Reviewer #2: Yes

4. Is the manuscript presented in an intelligible fashion and written in standard English?

Reviewer #1: Yes

Reviewer #2: Yes

5. Review Comments to the Author

Reviewer #1: This manuscript written by Dr. Matsukawa et al. aimed to clarify the effects of catecholamines in LPS-treated myotubes, and found that the epinephrine accelerates the LPS-induced muscle wasting and their related signaling pathway. With several types of pharmacological inhibition, the authors clarified the link between the two different signaling pathways of LPS and b-AR, in relation to IL-6 and STAT signaling. These experiments were well designed, and the presented results were convincing. However, it is still unclear why the epinephrine could enhance the LPS-induced muscle wasting, while the epinephrine treatment without LPS-induction had no effect on myotubes. The reviewer raised some concerns that need to be addressed in order to strengthen the conclusions drawn by the authors.

The critical concern was that it was not clear why epinephrine could accelerate the LPS-induced muscle wasting, and could activate relevant signal genes, namely STAT3, Atrogin-1 and NF-kB. One possibility is that the number of b-AR or the binding affinity to epinephrine were increased by LPS-treatment, and those augmentation enhanced the muscle wasting. Did the authors evaluate the expressions or the sensitivity of b-AR after LPS treatment? Did LPS-induced IL-6 affected them? Clarification of these would strongly support the hypothesis.

In Fig. 2, the authors evaluated the proteolysis related gene expressions including Atrogin-1 and MURF, but did not describe about the protein synthesis. Since the activation of b-AR increases skeletal muscle mass as previously described, it is a question whether the treatment with both LPS and epinephrine would affect the protein synthesis.

In Fig. 5, IL-6 inhibitor suppressed LPS-induced STAT3 activation and Atrogin-1 expression. The reviewer wonders if IL-6 inhibitor could prevent the myotube atrophy in vitro along with the alteration of these signaling molecules.

In addition, since NF-kB inhibitor also decreased these expressions, I wonder if the loss of MHC protein after LPS-treatment could be attenuated by Bay11-7082, as well.

In Fig. 6A and B, the results indicated that rIL-6 treatment (without LPS-induction) do not affect C/EBP� and Atrogin-1 expression even under the situation of p-STAT3 elevation, which implies that IL-6/STAT3 signaling is independent of C/EBPd-Atrogin1 signaling. However, in Fig. 9, the authors showed that C/EBPd was the downstream of IL-6/STAT3 pathway, which was inconsistent with other results shown in the manuscript. Explanations are required for these issues. In addition, please explain why LPS+IL-6 treatment could enhance C/EBPd-Atrogin1 signaling while rIL-6 did not affect these expressions.

In Fig. 6D and E, NF-kB inhibitor could abolish C/EBPd, Atrogin-1 and IL-6 expression as shown in Fig. 6 and 7, while it is still not clear whether this would affect the pSTAT3 signaling. Validation on the activation levels of STAT3 in Bay11-7082-treated myotubes is necessary.

Since epinephrine without LPS-treatment did not have any effects on myotube, it was unclear if b-AR mediated IL-6 signaling could directly affect Atrogin-1 expression and MHC protein amount as shown in Fig. 8. To answer this question,, experiments using b-AR overexpressed C2C12 might be required.

Reviewer #2: This paper is an exciting manuscript by Matsukawa et al. on sepsis's hot topic and its therapeutical strategies such as catecholamines. The idea of the rationale of the experiments is correct. However, the data showed is low-quality, especially those based in western blots that should contribute to the conclusions to a high degree. Thus, this reviewer thinks that the findings are not entirely based on the manuscript results.

Specific: the western blot of the Figures 1A, 2A, 2B, 5A do not represent the changes between LPS and LPS+ Epi depicted in the graphics. Also, Figure 6D does not show the differences between LPS and LPS+IL6. Why does LPS not increase the pSTAT3 in figure 5B?

Fluorescence microscopy in figure 1B presents high levels of background and complicated to see the myotubes delimitation.

6. PLOS authors have the option to publish the peer review history of their article (what does this mean?). If published, this will include your full peer review and any attached files.

Reviewer #1: No

Reviewer #2: No

---

## [Author Response · Author response to Decision Letter 0]

22 Apr 2021

Thank you for giving us the opportunity to strengthen our manuscript with your valuable comments and queries. We have incorporated changes that reflect the detailed suggestions you have graciously provided. Please see "Response to Reviewers" and "Manuscript" for the details. We also hope that our edits and the responses we provide below satisfactorily address all the issues and concerns you and the reviewers have noted.

---

## [Decision Letter · Decision Letter 1]

6 May 2021

Activation of the β-adrenergic receptor exacerbates lipopolysaccharide-induced wasting of skeletal muscle cells by increasing interleukin-6 production

PONE-D-21-02018R1

Dear Dr. Kai,

We’re pleased to inform you that your manuscript has been judged scientifically suitable for publication and will be formally accepted for publication once it meets all outstanding technical requirements.

Kind regards,

Atsushi Asakura, Ph.D

Academic Editor

PLOS ONE

Additional Editor Comments (optional):

Reviewers' comments:

Reviewer's Responses to Questions

**Comments to the Author**

1. If the authors have adequately addressed your comments raised in a previous round of review and you feel that this manuscript is now acceptable for publication, you may indicate that here to bypass the “Comments to the Author” section, enter your conflict of interest statement in the “Confidential to Editor” section, and submit your "Accept" recommendation.

Reviewer #1: All comments have been addressed

2. Is the manuscript technically sound, and do the data support the conclusions?

Reviewer #1: Yes

3. Has the statistical analysis been performed appropriately and rigorously? 

Reviewer #1: Yes

4. Have the authors made all data underlying the findings in their manuscript fully available?

Reviewer #1: Yes

5. Is the manuscript presented in an intelligible fashion and written in standard English?

Reviewer #1: Yes

6. Review Comments to the Author

Reviewer #1: (No Response)

7. PLOS authors have the option to publish the peer review history of their article (what does this mean?). If published, this will include your full peer review and any attached files.

Reviewer #1: No

---

## [Editor Report · Acceptance letter]

10 May 2021

PONE-D-21-02018R1 

Activation of the β-adrenergic receptor exacerbates lipopolysaccharide-induced wasting of skeletal muscle cells by increasing interleukin-6 production 

Dear Dr. Kai:

I'm pleased to inform you that your manuscript has been deemed suitable for publication in PLOS ONE. Congratulations! Your manuscript is now with our production department. 

Kind regards, 

on behalf of

Dr. Atsushi Asakura 

Academic Editor

PLOS ONE